# Enhancement of Water Uptake in Composite Superabsorbents Based on Carboxymethyl Cellulose Through Porogen Incorporation and Lyophilization

**DOI:** 10.3390/gels10120797

**Published:** 2024-12-05

**Authors:** Maria S. Lavlinskaya, Andrey V. Sorokin

**Affiliations:** 1Biophysics and Biotechnology Department, Voronezh State University, 1 Universitetskaya Square, 394018 Voronezh, Russia; sorokin_a@chem.vsu.ru; 2Polymer Science and Colloid Chemistry Department, Voronezh State University, 1 Universitetskaya Square, 394018 Voronezh, Russia

**Keywords:** composite superabsorbent, water uptake, swelling kinetics, porogens, lyophilization

## Abstract

Carboxymethyl cellulose sodium salt (CMC)-based superabsorbents are promising materials for the development of agricultural matrices aimed at water management and slow-release fertilizer production. However, an increase in the CMC content tends to reduce their water-absorbing capacity. This study aims to develop a cost-effective method for producing eco-friendly superabsorbents with enhanced water-absorbing capacity by incorporating a porogen and employing lyophilization. Superabsorbents containing 10 wt% CMC (CMC-SAPs) were synthesized via free radical polymerization with the addition of 0, 5, or 10 wt% ammonium carbonate as a porogen, followed by lyophilization. The synthesized CMC-SAPs were characterized using Fourier-transform infrared spectroscopy, scanning electron microscopy, thermogravimetric analysis, differential scanning calorimetry, and X-ray diffraction. The results revealed that CMC-SAPs prepared with the incorporation of a porogen and/or subjected to lyophilization exhibited well-developed surfaces featuring macropores and cavities. Incorporating 5 wt% ammonium carbonate as a porogen, followed by lyophilization, increased the equilibrium swelling ratio to 61%. This improvement was attributed to the enhanced surface morphology of the modified CMC-SAPs, which facilitated water molecule diffusion into the SAP matrix, as confirmed by open porosity measurements. This hypothesis was further supported by the diffusion coefficient values, which were higher for porogen-containing and lyophilized SAPs compared to unmodified samples. Moreover, the CMC-SAPs demonstrated good reusability. Thus, the combination of porogen incorporation and subsequent lyophilization represents a promising approach for enhancing the water uptake capacity of CMC-based composite superabsorbents for sustainable agricultural applications.

## 1. Introduction

New materials based on natural and synthetic polymers are increasingly being applied across various fields. For instance, hydrogels—hydrophilic, cross-linked polymers first developed in the 1950s—have found widespread use in biomedicine [1,2], food packaging [3,4], and contact lens production [5,6]. Among these, a specialized class of cross-linked hydrogels known as superabsorbents (SAPs) has garnered significant attention due to their ability to absorb large quantities of water.

Superabsorbents are extensively used in hygiene product manufacturing [7] and agricultural technologies [8]. Notably, their application in agriculture holds greater potential, as they not only function as water management systems but also serve as platforms for the controlled release of fertilizers [9,10,11].

Superabsorbents are typically synthesized from acrylic acid and its derivatives, such as salts, amides, and esters [12]. To enhance environmental sustainability, biodegradable components are often incorporated into the polymerization mixture, with various natural or modified polysaccharides being the most commonly utilized [13]. Materials containing both synthetic and natural segments are referred to as composite or hybrid superabsorbents.

Despite their improved eco-friendliness, composite SAPs generally exhibit lower water absorption compared to their fully synthetic counterparts. Furthermore, as the proportion of biodegradable components increases, the water-absorbing capacity of the composite SAP decreases [14,15,16,17]. This trade-off means that efforts to develop highly eco-friendly superabsorbents often result in products with diminished performance characteristics. Such limitations hinder the widespread adoption of composite superabsorbents in agricultural applications.

Various methods have been proposed to enhance the swelling performance of superabsorbents, typically characterized by the equilibrium swelling ratio. One approach involves reducing the overall rigidity of the polymer network. The most common strategy to achieve this is by decreasing the number of cross-links, often accomplished by lowering the amount of cross-linking agent used [18,19,20]. Another factor influencing network rigidity is the formation of intersegmental hydrogen bonds within the polymer matrix, which can occur between acrylate units, as well as polysaccharide segments, in superabsorbents containing carbohydrate components. To counteract the effects of these additional cross-links, the use of plasticizers has been suggested, as they can enhance the equilibrium swelling ratio [17]. A network rigidity decrease approach can be applied not only to composite SAPs but also to the synthetic ones. For instance, Qi and Hu [21] improved the polymer flexibility by substituting acrylamide—a common monomer in SAP synthesis—with (3-acrylamidopropyl)trimethylammonium chloride.

The rigidity of the polymer network is also significantly influenced by the nature of the cross-linking agent. Cross-linkers with higher functionality, i.e., more reactive centers for network formation, result in increased structural rigidity [21]. To address this, Hu et al. [22] proposed using 1,2,3,4-butane tetracarboxylic acid as a physical cross-linker. This compound forms hydrogen bonds with the hydroxyl groups of acrylic acid, reducing network rigidity. However, both this approach and a reduction in cross-linker concentration can compromise the mechanical strength of SAPs, which limits their practical applications.

Alternatively, Jiao et al. [23] incorporated porous CaCO_3_ nanoparticles into cross-linked acrylate SAPs to enhance the water absorbency without significantly affecting the mechanical properties. For polysaccharide-based superabsorbents, hydrophilization of the carbohydrate polymer has also been shown to improve the swelling performance [16].

Despite their efficacy, these methods often rely on industrially unavailable or expensive reagents and require additional processing steps, which increase the final product cost. This underscores the importance of developing cost-effective strategies to improve the water absorption capacity of SAPs that are practical for industrial applications.

An effective strategy for increasing the equilibrium swelling ratio of superabsorbents is to enhance their surface area through specific processing techniques. The most common method for creating polymeric materials with a developed surface involves the incorporation of pore-forming agents. One mechanism relies on the thermal decomposition of these agents, which releases gases that generate cavities and pores within the polymer matrix [24,25,26,27,28].

Another surface-enhancement approach is lyophilization, particularly suitable for hydrophilic materials [29]. Freeze-drying preserves the spatial conformation and arrangement of macromolecules after solvent removal, maintaining the structure of the dissolved or swollen material. Water, as an excellent solvent for hydrophilic polymers, allows the acrylate network to adopt a more expanded conformation, increasing the availability of sorption centers. Consequently, preserving the expanded conformation of the three-dimensional acrylate network is expected to significantly improve the equilibrium swelling ratio.

In this context, this study aims to develop a cost-effective method for eco-friendly producing carboxymethyl cellulose-based superabsorbents with enhanced water-absorbing capacity by incorporating a porogen and employing lyophilization.

## 2. Results and Discussion

### 2.1. Synthesis and Characterization of the Superabsorbents Based on Carboxymethyl Cellulose Sodium Salt

Superabsorbent polymers based on carboxymethyl cellulose sodium salt (CMC-SAPs) were successfully synthesized via free radical polymerization in an aqueous medium. The reactions were conducted using potassium persulfate as the initiator at 80 °C for 2 h. The synthesis parameters, including the comonomer ratio, polymerization feed concentration, and the amounts of initiator and cross-linker, were optimized based on our previous studies [16,17], which established conditions that maximize the equilibrium swelling ratio of CMC-SAPs.

The polymerization process is outlined in Figure 1. The mechanism follows the typical steps of a chain reaction and can be summarized as follows.

(i) Initiation: The thermal decomposition of potassium persulfate generates sulfate ion radicals, which react with the hydroxyl groups of CMC macromolecules to form macroradicals.

(ii) Propagation: These macroradicals subsequently react with acrylic acid, acrylamide, and N,N’-methylene-bis-acrylamide in a random sequence, leading to the growth of grafted polymer chains.

(iii) Termination: Recombination of the grafted polymer chains results in the formation of a three-dimensional polymeric network.

This stepwise process ensures the successful integration of CMC with synthetic monomers, resulting in a robust superabsorbent polymer matrix.

The structure of the synthesized CMC-SAPs was confirmed using Fourier-transform infrared (FTIR) spectroscopy. The FTIR spectrum (Figure 1) exhibits the following characteristic absorption bands: a broad band at 1051 cm^−1^, attributed to the symmetric stretching modes of the glucose ring skeletons and glycosidic bonds [30]; a band at 1117 cm^−1^, corresponding to the stretching of the acrylate -CO-O group [31]; a peak at 1400 cm⁻^1^, associated with the symmetric stretching of dissociated carboxylic groups [32]; bands at 1551 cm⁻^1^ and 1663 cm⁻^1^, assigned to the bending of NH₂ groups and the stretching of C=O groups, respectively [33]; a peak at 2932 cm⁻^1^, attributed to the asymmetric stretching of C–H bonds [31]; and a broad band in the range of 3200–3400 cm⁻^1^, indicating the vibrations of associated water molecules, as well as O-H and N-H stretching [17].

The FTIR spectra of the lyophilized SAPs and those containing ammonium carbonate were found to be similar, with no significant shifts or changes in the absorption bands. Therefore, only the FTIR spectrum of the CMC-SAP control sample is presented.

The surface morphology of CMC-SAPs was analyzed using scanning electron microscopy. The surface of CMC-SAPs synthesized without the inclusion of porogens or lyophilization appeared smooth, lacking noticeable cavities or pores (Figure 2a). In contrast, SAP samples prepared with ammonium carbonate exhibited a more developed surface, characterized by regular, sphere-like macropores. The number of pores increased with the higher ammonium carbonate content, consistent with the open porosity measurements, which showed an increase from 71% to 82% (Table 1 and Figure 2b,c).

Lyophilization significantly enhanced the surface roughness of the SAPs, producing irregular, faceted macropores with diameters exceeding those formed by ammonium carbonate alone (Figure 2d). Lyophilized SAPs containing ammonium carbonate exhibited a combination of sphere-like and faceted macropores, highlighting the synergistic effects of the two pore-forming techniques (Figure 2e,f). This combination also resulted in higher porosity values, reaching 93% and 95% for CMC-SAP-5AL and CMC-SAP-10AL, respectively.

Additionally, the surfaces of samples containing 10% (*w*/*w*) ammonium carbonate showed crystal-like deposits, attributed to excess ammonium carbonate remaining after synthesis.

The thermal stability of polymers is a critical factor in assessing their suitability for practical applications. To evaluate this property in the synthesized CMC-SAPs, thermogravimetric analysis (TGA) was performed in an inert helium atmosphere over a temperature range of 25–580 °C (Figure 3). The TGA curves for all CMC-SAP samples exhibited similar trends, with three distinct stages of weight loss identified.

The first stage occurred below 315 °C, with weight losses ranging from 19% to 26.4%, attributed to the evaporation of water associated with the CMC-SAPs. The second stage, observed between 315 °C and 400 °C, corresponded to the decomposition of the grafted polyacrylic segments, leaving residual masses between 54.7% and 63.6%. The third stage, occurring above 400 °C, was associated with the decomposition of the CMC backbone and the decarboxylation of previously formed anhydride groups [34]. The final residual masses ranged from 33.5% to 49.5%.

It was observed that the introduction of porosity did not significantly impact the thermal stability of the CMC-SAPs, as the total differences in weight loss among the samples were generally less than 10%. An exception was noted for CMC-SAP-10AL, which exhibited a final weight loss difference of approximately 16%.

The addition of porogens and the process of lyophilization can significantly influence the internal structure of superabsorbents. One critical parameter that reflects the interactions within a three-dimensional network is the glass transition temperature (*T_g_*). Higher *T_g_* values indicate stronger intersegmental interactions and increased rigidity of the polymer network, which can impact the swelling capacity of superabsorbents in water. Our previous research [17] demonstrated that reducing *T_g_* values can enhance the equilibrium swelling ratio. In this study, the glass transition temperatures of the synthesized CMC-SAPs were measured using differential scanning calorimetry, with the results summarized in Table 1 and Appendix A. The data reveal an increase in *T_g_* values compared to the control CMC-SAP sample. Notably, despite this increase, the equilibrium swelling ratios of the CMC-SAP-A and CMC-SAP-AL samples exceeded that of the control CMC-SAP.

These findings suggest that the improved water-absorbing capacity of the composite CMC-SAPs is not primarily driven by a reduction in intersegmental interactions. Instead, it can be attributed to the development of a larger polymer surface area, which enhances interactions with water molecules.

### 2.2. Investigation of Water Sorption

The equilibrium swelling ratio (*Q_e_*) is a critical property of superabsorbents that determines their suitability for various applications. As previously discussed, the degree of cross-linking and the rigidity of the polymer network play a significant role in influencing the swelling behavior of SAPs by restricting the availability of the sorption centers. Although reducing these parameters can improve *Q_e_*, it may also adversely affect other essential properties of SAPs, such as mechanical strength and rheological behavior.

An alternative strategy to increase the availability of sorption centers is to enhance the polymer’s surface area. In this study, ammonium carbonate is proposed as a porogen, either independently or in combination with lyophilization, to improve the equilibrium swelling ratio.

The equilibrium swelling ratio (*Q_e_*) values are summarized in Table 1. The results indicate that the addition of a porogen and the use of lyophilization significantly enhance the *Q_e_* of the synthesized SAPs. The highest *Q_e_* value was observed for CMC-SAP-5AL, while the lowest was recorded for CMC-SAP-10A. Nevertheless, even CMC-SAP-10A exhibited a *Q_e_* value 9% higher than that of the blank CMC-SAP. Additionally, SAPs incorporating ammonium carbonate and subjected to lyophilization displayed higher *Q_e_* values than the lyophilized-only sample (CMC-SAP-L), suggesting a synergistic effect between the pore-forming factors. Furthermore, CMC-SAP-5A showed a higher *Q_e_* compared to CMC-SAP-10A, and a similar trend was noted for CMC-SAP-5AL versus CMC-SAP-10AL. Notably, there was no direct correlation between SAP porosity (Table 1) and *Q_e_* values. This discrepancy appears to arise from the excess ammonium carbonate. SEM images reveal crystal-like deposits on the surface of SAPs containing 10% ammonium carbonate alongside the pores (Figure 2c,f). These deposits correspond to excess salt adsorbed on the polymer surface. When immersed in water, the ammonium carbonate dissolves, altering the solution’s ionic strength and impeding the swelling of the SAPs.

An analysis of previously published data suggests that the molecular weight (MW) of carbohydrates significantly influences the equilibrium swelling ratio (*Q_e_*) of superabsorbents [35]. In a prior study [17], we demonstrated that a CMC-SAP containing 10% *w*/*w* carboxymethyl cellulose (CMC) with a MW of 10,000 and a degree of substitution (DS) of 0.72 exhibited a *Q_e_* value of 1061 ± 51. In the present study, CMC with a MW of 90,000 and a DS of 0.7 were used, resulting in a *Q_e_* value of 878 ± 32. Since the DS values of the two CMC samples are comparable and the SAP syntheses were conducted under identical conditions, this comparison is valid. These results indicate that the equilibrium swelling ratio decreases as the molecular weight of CMC increases. Notably, these findings align with previously reported data. For instance, in [15], the effect of chitosan molecular weight on water absorption was investigated, demonstrating that, similar to our findings, *Q_e_* values decrease as the molecular weight of the polysaccharide increases.

Additionally, structural differences in the CMC samples appear to contribute to these variations. The food-grade CMC with a MW of 10,000, intended for use as a thickener, is highly amorphous, as evidenced by the appearance of a broad peak at 20.2° in its X-ray diffraction (XRD) pattern (Figure 4). In contrast, the CMC with a MW of 90,000 exhibits greater crystallinity, confirmed by XRD peaks characteristic of the cellulose Iβ structure [36]. These structural distinctions further influence the SAP equilibrium swelling ratio.

The medium in which SAP swelling occurs has a significant impact on the process. It is well established that the ionic strength of a solution dramatically reduces the swelling ratio due to the effects of osmotic pressure and ion screening [37]. To examine the influence of salinity on the equilibrium swelling ratio, experiments were performed in a 0.15 M NaCl solution (Table 1). The results reveal a marked decrease in *Q_e_* values for CMC-SAPs in the presence of electrolytes. This reduction can be attributed to the additional effect of cations, which diminish electrostatic repulsion within the polymer network. This, in turn, leads to a decrease in the Donnan osmotic pressure difference between the polymer network and the external solution. The reduced difference in mobile ion concentration between the polymer network and the surrounding liquid further lowers the absorbency capacity [37].

Nevertheless, the overall trend observed in deionized water is maintained: porous SAPs consistently exhibit higher equilibrium swelling ratios than their non-porous counterparts, even in saline conditions.

Structural modifications in superabsorbents can significantly affect the swelling behavior of their three-dimensional networks. The swelling kinetics of the CMC-SAPs are presented in Figure 5a. The data reveal that, although the swelling curves are similar across all samples, CMC-SAPs containing porogens or subjected to lyophilization exhibit a faster swelling rate, as evidenced by a steeper initial rise in the curves.

All curves display a consistent overall profile with two distinct phases: an initial rapid swelling phase occurring within the first 1–30 min, followed by a slower swelling phase from 30 to 120 min. After this period, equilibrium is reached, and the swelling ratio stabilizes.

To explore the mechanism of superabsorbent interactions with water, the swelling kinetics were analyzed using various mathematical models, including the pseudo-first-order, pseudo-second-order, and Ritger-Peppas models. The processed data are shown in Figure 5b–d. For this analysis, two samples were selected: CMC-SAP and CMC-SAP-5AL, the latter demonstrating the highest equilibrium swelling ratio. The suitability of each model was evaluated based on the R^2^ coefficient and the non-parametric chi-square test.

As shown in Table 2, the pseudo-second-order model provided the best fit for both samples (CMC-SAP: χ^2^ = 1 << χ_c_^2^ = 15.5; CMC-SAP-5AL: χ^2^ = 0.8 << χ_c_^2^ = 15.5). This model is based on the chemisorption mechanism, which involves interactions with the sorbate through the formation of various types of bonds, such as hydrogen bonds in this case. Furthermore, the rate constants *k*_1_ and *k*_2_, derived from the graphical interpretations of the pseudo-first-order and pseudo-second-order models, were higher for CMC-SAP-5AL compared to CMC-SAP (Table 2).

It is well known that the swelling behavior of SAP is highly dependent on the SAP-to-water ratio, particularly the volumetric effect. To investigate the swelling characteristics, three different SAP-to-water ratios were analyzed: 0.1 g SAP per 500 mL of water, 0.5 g SAP per 500 mL of water, and 1.0 g SAP per 500 mL of water. The results (Appendix A) indicate that an increase in CMC-SAP mass leads to a decrease in *Q_e_* values. However, the swelling kinetic profiles exhibit similar shapes across all conditions. The pseudo-second-order model remains the most suitable for describing the data, suggesting that chemisorption is the driving force behind the swelling in all the cases studied.

Chemisorption is not the only factor influencing the swelling behavior of superabsorbents; the diffusion mechanism of water molecules within the polymer matrix also plays a crucial role. This diffusion mechanism can be characterized by calculating the nnn-parameter from the Ritger-Peppas model equation. When *n* = 0.5, the diffusion is classified as Case I or Fickian diffusion, where the rate of water diffusion is slower than the polymer relaxation rate. For *n* = 1, the process corresponds to Case II or non-Fickian diffusion, where the water diffusion rate surpasses the polymer relaxation rate. When 0.5 < *n* < 1, the diffusion is classified as Case III or anomalous diffusion, a non-Fickian process where the water diffusion and polymer relaxation rates are approximately equal. Additionally, *n* values outside this range have been reported, with *n* > 1 indicating super Case II diffusion and *n* < 0.5 corresponding to pseudo-Fickian diffusion [38,39]. Therefore, the diffusion type is largely determined by the intrinsic properties of the polymers.

Figure 5d presents the *n*-parameter values obtained from fitting the data to the Ritger-Peppas model. For both SAPs, *n* < 0.5, indicating pseudo-Fickian diffusion, a behavior characteristic of highly ionized polymers [40]. However, the corresponding R^2^ values were relatively low (0.908 for CMC-SAP and 0.951 for CMC-SAP-10AL), suggesting the need for an alternative analytical approach. To address this, it is proposed to separately analyze the Ritger-Peppas model data for the fast and slow sorption stages (Figure 5e).

For CMC-SAP, *n* < 0.5, whereas, for CMC-SAP-5AL, *n* > 0.5, indicating a shift in the diffusion mechanism from pseudo-Fickian to non-Fickian diffusion, likely due to an increase in the surface area of the SAPs. During the slow water sorption stage, both SAPs exhibited *n* < 0.5. Additionally, the R^2^ values for the fast and slow sorption stages of CMC-SAP-5AL were relatively high, exceeding 0.96 in both cases. The chi-square value (χ^2^ = 0.9) was also significantly lower than the critical value (χ_c_^2^ = 15.5), confirming the reliability of the results.

The diffusion rate within the SAP matrix can be quantified using the diffusion coefficient. The diffusion coefficients (Figure 5f; Table 2) were calculated using the short-time approximation method, which is valid for the initial 60% of the swelling process [34,41]. The results indicate that the diffusion coefficient values for CMC-SAP-5AL are higher than those for CMC-SAP. This finding suggests that the porous structure of CMC-SAP-5AL not only enhances the accessibility of sorption sites for interactions with water molecules but also facilitates the more rapid transport of water molecules to these active sites.

In the context of practical agricultural applications of SAPs, reusability is a critical factor, as soil undergoes multiple humidity fluctuations. Figure 6 illustrates the behavior of CMC-SAP during five swelling–deswelling cycles. Notably, after five cycles, both SAPs retained more than 85% of their initial equilibrium swelling ratio, indicating high stability of the polymer networks and highlighting their potential for industrial applications.

Carboxymethyl cellulose-based superabsorbents are extensively studied due to the abundance and low cost of this modified polysaccharide, as well as their high potential for industrial applications. Various techniques have been employed to fabricate the spatial structures of these materials. Table 3 summarizes the research efforts over the past decade on the production of CMC-based SAPs with varying compositions. As shown, the equilibrium swelling ratio is highly dependent on the SAP composition, underscoring the ongoing need to explore innovative approaches for producing low-cost, eco-friendly materials with high water-absorbing capacities.

## 3. Conclusions

In this study, we demonstrated an easy and cost-effective method for producing eco-friendly CMC-based superabsorbents with enhanced water-absorbing capacity. CMC-based superabsorbent polymers containing 10 wt% CMC were successfully synthesized via free solution polymerization. The use of sodium carboxymethyl cellulose with a higher molecular weight and a more ordered structure led to a decrease in the equilibrium swelling ratio. However, incorporating 5 wt% ammonium carbonate and applying lyophilization increased the *Q_e_* value of the synthesized SAPs by up to 61%. This improvement is attributed to the enhanced surface development of the CMC-SAPs, while intra-network interactions remained largely unchanged in the presence of a porogen or under lyophilization conditions.

The diffusion coefficient analysis revealed that surface development contributes to its increase, facilitating improved swelling in both distilled water and saline solutions. Based on these findings, the combination of porogen incorporation and lyophilization emerges as a promising and effective strategy for enhancing the equilibrium swelling ratio of composite superabsorbents. The resulting products show great potential in agriculture as matrices for slow-release fertilizers and water management applications. These aspects will be further investigated and developed in our future work.

## 4. Materials and Methods

### 4.1. Materials

Carboxymethyl cellulose sodium salt (CMC; >95%), with an average molecular weight (Mw) of 90,000, a degree of substitution of 0.7, and a viscosity of 110 mPa·s for a 2% aqueous solution at 25 °C, was purchased from FGUP KKhK, Moscow, Russia, and used for superabsorbent polymer (SAP) synthesis without further purification. Acrylamide (AAm; extra pure, >98%; Acros Organics, Geel, Antwerpen, Belgium), acrylic acid (AA; extra pure, >98%; Acros Organics, Geel, Antwerpen, Belgium), *N*,*N*′-methylene-*bis*-acrylamide (MBAAm; >98%; Acros Organics, Geel, Antwerpen, Belgium), potassium persulfate (PPS; >98%; Acros Organics, Geel, Antwerpen, Belgium), and potassium hydroxide (>98%; Acros Organics, Geel, Antwerpen, Belgium) were used in the SAP synthesis. Prior to use, AA was distilled under vacuum (boiling point = 45 °C/15 mmHg), AAm and MBAAm were recrystallized from ethyl acetate, and PPS was recrystallized from deionized water. Ammonium carbonate (>98%), obtained from Vekton, Saint Petersburg, Russia, was used as a porogen. Distilled water (21 MΩ, pH 6.7 ± 0.2), ethanol (anhydrous, 99.5%, ReaKhim, Moscow, Russia), and ethyl acetate (chromatography grade, >99%, ReaKhim, Moscow, Russia) were used as solvents without further treatment.

### 4.2. Superabsorbent Synthesis

The CMC-SAPs were synthesized following a previously described procedure [17], with minor modifications. The synthesis protocol remained the same for CMC-SAP-10-0; however, 5 wt% or 10 wt% of ammonium carbonate was added to the reactor after the introduction of the acrylic comonomers. Following the synthesis, the samples were lyophilized to a constant weight after ethanol treatment. The resulting superabsorbents were designated as CMC-SAP-5A and CMC-SAP-10A, respectively. For samples containing ammonium carbonate and subsequently lyophilized, the designations were CMC-SAP-5AL and CMC-SAP-10AL, while the sample that was only lyophilized was denoted as CMC-SAP-10L. The yield of the products after freeze-drying ranged from 88% to 93%. The synthesized CMC-SAPs were found to be insoluble in water, ethanol, isopropanol, and acetone.

### 4.3. Instrumental Characterization

#### 4.3.1. Fourier-Transform Infrared Spectroscopy

Fourier-transform infrared spectroscopy (FTIR) with attenuated total reflectance (ATR) was used for the structural characterization of CMC-SAP. Spectra were recorded using a Bruker Vertex 70 instrument (Bruker Corporation, Billerica, MA, USA) with a Fourier transducer in the 400–4000 cm⁻^1^ range. Each measurement consisted of 32 scans per cycle, with a total of 4 cycles. The samples were analyzed in powder form.

#### 4.3.2. Scanning Electron Microscopy

Scanning electron microscopy (SEM) was employed to examine the surface morphology of the CMC-SAP. Images were acquired using a JEOL JSM-6380LV scanning electron microscope (JEOL Ltd., Tokyo, Japan) in secondary electron imaging (SEI) mode. Prior to analysis, the samples were sputter-coated with a 10 nm thick gold layer.

#### 4.3.3. Thermogravimetric Analysis and Differential Scanning Calorimetry

Thermogravimetric analysis (TGA) and differential scanning calorimetry (DSC) were used to confirm the structural properties and determine the glass transition temperature (T_g_) of the CMC-SAP. The measurements were carried out on a STA 449 F3 Jupiter simultaneous thermal analyzer (Erich Netzsch B.V & Co., Holding KG, Selb, Germany). The analysis was performed in capped aluminum pans under a helium atmosphere, with a heating rate of 10 °C/min over a temperature range of 20 °C to 600 °C.

#### 4.3.4. X-ray Diffraction

X-ray diffraction (XRD) was used to assess the crystallinity of the CMC samples. The analysis was conducted using an Empyrean diffractometer (Malvern Panalytical B.V., Almelo, The Netherlands) equipped with a Cu-Kα radiation source (λ = 1.54 Å, 45 kV, 35 mA). Measurements were performed over a scattering angle (2θ) range of 10° to 80°, with a step size of 0.02° and a scanning speed of 2°/min.

### 4.4. Porosity Evaluation

The open porosity (*P*, %) of the CMC-SAPs was determined using the liquid displacement method. A sample was submerged in a known initial volume of ethanol (*V*_1_), and a series of brief evacuation–repressurization cycles were performed to ensure the liquid fully penetrated the sample’s pores. The volume of the ethanol-saturated sample was recorded as *V*_2_. After removing the liquid-impregnated CMC-SAP, the remaining liquid volume was measured as *V*_3_. The open porosity was then calculated as described in [49]:(1)$$P=\frac{V_{1}-V_{3}}{V_{2}-V_{3}}.$$

### 4.5. Swelling Research

The equilibrium swelling ratio and swelling kinetics in distilled water or a 0.15 M NaCl aqueous solution were determined following the method described by Sorokin and Lavlinskaya [16]. The equilibrium swelling ratio, *Q_e_*, and the swelling ratio at time *t*, *Q_t_*, are calculated as
(2)$$Q=\frac{m_{1}-m_{0}}{m_{0}},$$
where *m*_1_ and *m*_0_ are the weights of the swollen and dry CMC-SAP samples, respectively. Three samples were tested for each synthesized SAP (*n* = 3, *P* = 0.95). The results are presented as mean values ± standard deviation based on three independent experiments, analyzed using MS Excel software.

To investigate the water absorption mechanism and the influence of a pore forming, the swelling kinetics data were fitted to standard mathematical kinetic swelling models. The pseudo-first-order swelling kinetic model is represented as [50]
(3)$$\ln\left(Q_{e}-Q_{t}\right)=\ln Q_{e}-k_{1}t\;,$$
the pseudo-second-order swelling kinetic model is described by the following equations [51]:(4)$$\frac{t}{Q_{t}}=\frac{1}{k_{2}Q_{e}^{2}}+\frac{t}{Q_{e}},$$
and the Ritger-Peppas model is characterized as [52]
(5)$$F=\frac{Q_{t}}{Q_{e}}=k\times t^{n},$$
or the logarithm Expression (4):(6)$$\ln F=\ln Q_{t}-\ln Q_{e}=\ln k+n\ln t,$$
where *Q_e_* (g/g) and *Q_t_* (g/g) represent the swelling ratios at equilibrium and at time t, respectively; *k*_1_ and *k*_2_ (g/mg min) are the rate constants for the pseudo-first-order and pseudo-second-order models, respectively; F denotes the fractional swelling rate at time *t*; *k* is the structural parameter; and *n* is the swelling exponent indicating the type of diffusion mechanism. To confirm the results, a chi-square (χ^2^) non-parametric test was performed with a significance level of 0.05 and degrees of freedom equal to 9 (χ_c_^2^ = 15.5).

For determining the diffusion coefficient, a short-time approximation method was applied [34], which is valid for the initial 60% of the equilibrium swelling ratio. For spherical SAP particles, the diffusion coefficient, *D*, is expressed by the following equation [52]:(7)$$\frac{Q_{t}}{Q_{e}}=4\sqrt{\frac{Dt}{\pi r^{2}}}$$
where *D* (cm^2^/min) is the diffusion coefficient, *t* (min) is time, and *r* (cm) is particle radius. The slope of the *Q_t_*/*Q_e_* versus *t*^1/2^ plot corresponds to the diffusion coefficient value.

## Data Availability

The original contributions presented in the study are included in the article/Appendix A, and further inquiries can be directed to the corresponding author.

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
