# Peer review of "Enhancement of Water Uptake in Composite Superabsorbents Based on Carboxymethyl Cellulose Through Porogen Incorporation and Lyophilization"

_gels, 2024, doi:10.3390/gels10120797_

Round 1
Reviewer 1 Report
Comments and Suggestions for Authors
This manuscript examines the impact of porogenic ammonium carbonate and the lyophilization method on the water absorbency of Carboxymethyl Cellulose-based superabsorbents. The text encompasses the swelling characteristics of Carboxymethyl Cellulose-based superabsorbent polymer (Carboxymethyl Cellulose SAP) in water and a 0.15 M NaCl electrolyte. Super absorbents comprising 10% CMC (CMC-SAPs) by weight were produced via free radical polymerization, incorporating 0, 5, or 10% ammonium carbonate by weight. Subsequently, lyophilization and swelling equilibrium kinetics investigations were performed. This method innovatively imparts porogenic features to Carboxymethyl Cellulose-based superabsorbents by the incorporation of ammonium carbonate and enhances swelling characteristics via lyophilization. The manuscript references appropriate literature. Only the literature from the 25th is in Russian. The English edition of the 25th literature is inaccessible. If an English translation exists, it may be favored in the manuscript.
Figures and tables are appropriate.
This manuscript is acceptable with minor revisions.
My contributions are outlined below.
1. The part about superabsorbent synthesis needs greater elaboration.
The action of NaCl as an electrolyte in Page 7 of paragraph 2 should be articulated with clarity, as demonstrated in the 26th reference cited in the manuscript.
2. In the NaCl solution, the swelling ratio of all hydrogel samples primarily decreased as the salt content increased. The expansion and contraction of hydrogels in saline solutions are influenced by the ionic interactions between mobile ions and fixed charges, significantly affecting the osmotic pressure between the hydrogel's interior and the exterior solution.At elevated Na+ concentrations, the gels commenced shrinking due to a reduction in Donnan osmotic pressure. Furthermore, hydrogels with varying chemical compositions demonstrated distinct shrinkage behaviors, attributable to the charge density inside the hydrogel networks, necessitating a more extensive examination based on existing research.
3. It is denoted as superabsorbent polymers (SAPs) in both explicit and abbreviated forms. Given the prior lengthy and concise references, it is more suitable to use SAP in this and the subsequent phrases.
(line 156).
4. The manuscript includes XRD, although it is not referenced in the abstract.
5. DSC is mentioned, but there is no DSC thermogram in the Manuscript.
6. No grammatical errors were identified in English; however, the language should be rendered more fluent.
Comments on the Quality of English LanguageNo grammatical errors were identified in English; however, the language should be rendered more fluent.
Author Response
We would like to thank the Reviewer for careful reading of our manuscript. In the view of the constructive criticism by the Reviewer, we have revised the manuscript considerably. All our corrections are highlighted in yellow in the text.
This manuscript examines the impact of porogenic ammonium carbonate and the lyophilization method on the water absorbency of Carboxymethyl Cellulose-based superabsorbents. The text encompasses the swelling characteristics of Carboxymethyl Cellulose-based superabsorbent polymer (Carboxymethyl Cellulose SAP) in water and a 0.15 M NaCl electrolyte. Super absorbents comprising 10% CMC (CMC-SAPs) by weight were produced via free radical polymerization, incorporating 0, 5, or 10% ammonium carbonate by weight. Subsequently, lyophilization and swelling equilibrium kinetics investigations were performed. This method innovatively imparts porogenic features to Carboxymethyl Cellulose-based superabsorbents by the incorporation of ammonium carbonate and enhances swelling characteristics via lyophilization. The manuscript references appropriate literature.
Figures and tables are appropriate.
This manuscript is acceptable with minor revisions.
My contributions are outlined below.
Point 1. The part about superabsorbent synthesis needs greater elaboration. The action of NaCl as an electrolyte in Page 7 of paragraph 2 should be articulated with clarity, as demonstrated in the 26th reference cited in the manuscript.
Response 1. A detailed discussion of the synthesis is presented in our earlier published work (Ref. 17). To avoid repetition, we have included a synthesis flowchart (Scheme 1). The description of the swelling results in saline solution has been revised accordingly.
Point 2. In the NaCl solution, the swelling ratio of all hydrogel samples primarily decreased as the salt content increased. The expansion and contraction of hydrogels in saline solutions are influenced by the ionic interactions between mobile ions and fixed charges, significantly affecting the osmotic pressure between the hydrogel's interior and the exterior solution. At elevated Na+ concentrations, the gels commenced shrinking due to a reduction in Donnan osmotic pressure. Furthermore, hydrogels with varying chemical compositions demonstrated distinct shrinkage behaviors, attributable to the charge density inside the hydrogel networks, necessitating a more extensive examination based on existing research.
Response 2. Thank you for your valuable suggestion. The low swelling degree of anionic hydrogels in saline media is indeed a significant challenge for these materials, limiting the application of SAPs. Therefore, studying the swelling behavior of anionic SAPs with different compositions in media of varying ionic strength is crucial for gaining a fundamental understanding of SAP interaction mechanisms. However, in this study, we focused on SAPs with the same composition, differing only in post-synthetic treatment.
Your comment has inspired us to plan a new study in which we will compare the swelling behavior of CMC- and chitosan-based SAPs in different saline media. We hope that our future work will be of interest to you and will be reviewed by you
Point 3. It is denoted as superabsorbent polymers (SAPs) in both explicit and abbreviated forms. Given the prior lengthy and concise references, it is more suitable to use SAP in this and the subsequent phrases (line 156).
Response 3. This sentence has been modified.
Point 4. The manuscript includes XRD, although it is not referenced in the abstract.
Response 4. We have included a mention of XRD in the abstract.
Point 5. DSC is mentioned, but there is no DSC thermogram in the Manuscript.
Response 5. The DSC profiles have been added to the manuscript as Fig. S1.
Point 6. No grammatical errors were identified in English; however, the language should be rendered more fluent.
Response 6. We have made efforts to improve the English. We hope you find it suitable.
Point 7. Only the literature from the 25th is in Russian. The English edition of the 25th literature is inaccessible. If an English translation exists, it may be favored in the manuscript.
Response 7. Unfortunately, the only abstract is available in English. Now this ref. is 15.
Thanks for your comments. You help us become better!
Reviewer 2 Report
Comments and Suggestions for Authors
Before publishing in gels, the authors need to address the following points regarding their work on Enhancement of Water Uptake in Composite Superabsorbents Based on Carboxymethyl Cellulose through Porogen Incorporation and Lyophilization.
1. Ensure that your abstract clearly highlights the objectives of your study, the materials and methods used the results and discussion, and the novelty of your work.
2. Provide a comprehensive literature review in your introduction, starting with general information and narrowing down to the specific research gap your work fills. Highlight how your work serves the needs of a broad audience. The current introduction lacking significant connectivity between the sentences.
3. It is recommended to kindly add a flowchart that best describes the synthesis process used in this study.
4. Figure 1 requires revisions. The unit of absorbance is omitted and should be included. Moreover, the FTIR spectrum is presented in reverse order and deviates from the standard format. To improve readability, the x-axis label should state the full variable name. Finally, instead of citing references at the end, incorporate specific citations alongside each peak assignment to facilitate direct correlation with relevant literature.
5. Figure 2, comprising images (a-f), lacks detailed explanations for each SEM image. On page 3 (lines 111-119), the authors provide a general description without explicitly referencing the corresponding figures. To enhance clarity, it is suggested that the authors provide a figure-by-figure explanation, specifically referencing each image (a-f), to facilitate a deeper understanding of the SEM imaging results.
6. TGA figure requires additional information to improve clarity. The authors should add the quantity of weight loss (%) corresponding to each of the three stages, complementing the already mentioned temperature ranges. Currently, only temperatures are provided, lacking explicit percentage values for weight loss at each stage.
7. Figure 4 lacks proper explanation in the manuscript
8. The suitability of the models was assessed through R-squared coefficient. In general, this parameter is not reliable. It is recommended to use some other statistical metrics that shows the efficiency of the applied models.
9. In the diffusion discussion none of the model best fits the experimental data, other than the pseudo second order model that cross pond to the chemisorption phenomena. It was well established in the literature that the swelling kinetics for hybrid hydrogels containing filler with charge moites or functional groups were followed pseudo second order kinetics. Therefore, it is important to analyses the effect of concentration on swelling kinetics to verify any deviation of swelling kinetics from this obvious behavior.
10. According to literature, the best correlation is established when k2 > k1. In the current study it is vice versa and still R2 is greater for second order. Can you explain this discrepancy?
11. The diffusion type Fickian and non Fickian were predicted using Ritger-Peppas model. However, diffusion data clearly demonstrate in figure (5e) page 8 that the model was not well fit and the predicted slope n was based on the fitting curve of two data points, especially at initial sorption stage (lower swelling ratio/ lower time). This slope or n value could be different if the authors omitted the initial data point (startup effect). It is recommended to reevaluate the data for correct prediction of diffusion type.
12. It is recommended to show connected lines with symbol in figure 5a for better readability of the swelling behaviors. This figure shows that the data point was not observed at the same time intervals during the swelling experiment. Higher data points were shown at initial swelling regime. This variation in the swelling measurement could significantly affect the diffusion kinetics and diffusion type prediction as well as does not specify the exact time of change in swelling behavior. Author needs to explain this variance in the measurement.
13. What are the future developments of your work??
14. To provide a comprehensive overview of the research landscape, it is recommended that the authors include a table summarizing the work of other researchers in this area. This table should compare and contrast the key findings, methods, and materials used in similar studies, allowing readers to easily visualize the contributions and advancements in the field.
15. Begin your conclusion section with a summary of your work, emphasizing the main findings and the gap your work fills in the literature. Conclude your work in this section as well. Current version only summarizes your work without concluding
Author Response
We would like to thank the Reviewer for careful reading of our manuscript. In the view of the constructive criticism by the Reviewer, we have revised the manuscript considerably. All our corrections are highlighted in yellow in the text.
Before publishing in gels, the authors need to address the following points regarding their work on Enhancement of Water Uptake in Composite Superabsorbents Based on Carboxymethyl Cellulose through Porogen Incorporation and Lyophilization.
Point 1. Ensure that your abstract clearly highlights the objectives of your study, the materials and methods used the results and discussion, and the novelty of your work.
Response 1. We have modified the abstract according to your suggestions.
Point 2. Provide a comprehensive literature review in your introduction, starting with general information and narrowing down to the specific research gap your work fills. Highlight how your work serves the needs of a broad audience. The current introduction lacking significant connectivity between the sentences.
Response 2. We have made changes to the introduction based on your suggestions. We hope you find them suitable.
Point 3. It is recommended to kindly add a flowchart that best describes the synthesis process used in this study.
Response 3. The synthesis flowchart has been added as Scheme 1.
Point 4. Figure 1 requires revisions. The unit of absorbance is omitted and should be included. Moreover, the FTIR spectrum is presented in reverse order and deviates from the standard format. To improve readability, the x-axis label should state the full variable name. Finally, instead of citing references at the end, incorporate specific citations alongside each peak assignment to facilitate direct correlation with relevant literature.
Response 4. Fig. 1 (FTIR spectrum) has been modified according to your comment. References describing the FTIR bands have also been added.
Point 5. Figure 2, comprising images (a-f), lacks detailed explanations for each SEM image. On page 3 (lines 111-119), the authors provide a general description without explicitly referencing the corresponding figures. To enhance clarity, it is suggested that the authors provide a figure-by-figure explanation, specifically referencing each image (a-f), to facilitate a deeper understanding of the SEM imaging results.
Response 5. We have made efforts to enhance the SEM descriptions. Additionally, we have compared these results with porosity data of the superabsorbents, evaluated using the displacement method.
Point 6. TGA figure requires additional information to improve clarity. The authors should add the quantity of weight loss (%) corresponding to each of the three stages, complementing the already mentioned temperature ranges. Currently, only temperatures are provided, lacking explicit percentage values for weight loss at each stage.
Response 6. Data on the weight loss percentage have been added to the TGA paragraph.
Point 7. Figure 4 lacks proper explanation in the manuscript
Response 7. The interpretation of Fig. 4 has been added to the manuscript.
Point 8. The suitability of the models was assessed through R-squared coefficient. In general, this parameter is not reliable. It is recommended to use some other statistical metrics that shows the efficiency of the applied models.
Response 8. In response to your comment, a non-parametric chi-square test has also been applied to determine the best-fitted kinetic model.
Point 9. In the diffusion discussion none of the model best fits the experimental data, other than the pseudo second order model that cross pond to the chemisorption phenomena. It was well established in the literature that the swelling kinetics for hybrid hydrogels containing filler with charge moites or functional groups were followed pseudo second order kinetics. Therefore, it is important to analyses the effect of concentration on swelling kinetics to verify any deviation of swelling kinetics from this obvious behavior.
Response 9. The concentration dependency of the swelling kinetics of CMC-SAPs is presented in Fig. S2.
Point 10. According to literature, the best correlation is established when k2 > k1. In the current study it is vice versa and still R2 is greater for second order. Can you explain this discrepancy?
Response 10. Certainly, when k2 > k1, better correlation is achieved. However, there are many examples of sorption of various substances where the opposite is observed, yet the pseudo-second-order model remains more appropriate (https://link.springer.com/article/10.1007/s10924-020-01720-z/ https://www.scirp.org/journal/paperinformation?paperid=2969 https://www.mdpi.com/2073-4360/14/23/5175 https://www.sciencedirect.com/science/article/pii/S1876107023001591 ).
Point 11. The diffusion type Fickian and non Fickian were predicted using Ritger-Peppas model. However, diffusion data clearly demonstrate in figure (5e) page 8 that the model was not well fit and the predicted slope n was based on the fitting curve of two data points, especially at initial sorption stage (lower swelling ratio/ lower time). This slope or n value could be different if the authors omitted the initial data point (startup effect). It is recommended to reevaluate the data for correct prediction of diffusion type.
Response 11. Following your suggestion, we recalculated the n values by omitting the first experimental data. Fig. 5e has also been modified accordingly. However, the results remain the same.
Point 12. It is recommended to show connected lines with symbol in figure 5a for better readability of the swelling behaviors. This figure shows that the data point was not observed at the same time intervals during the swelling experiment. Higher data points were shown at initial swelling regime. This variation in the swelling measurement could significantly affect the diffusion kinetics and diffusion type prediction as well as does not specify the exact time of change in swelling behavior. Author needs to explain this variance in the measurement.
Response 12. Fig. 5a has been modified according to your suggestion. The time intervals were selected based on following. The SAPs were crushed, and the 50 μm fraction was used in the research. Observations indicate that the most significant changes occur in the first 30 minutes of swelling. Since the majority of data points were obtained during this time interval, while the end of the experiment corresponds to the equilibrium state being reached.
Point 13. What are the future developments of your work??
Response 13. This paper is a part of continuous research. We plan to test the developed CMC-SAPs in agriculture as matrices for slow-release fertilizers.
Point 14. To provide a comprehensive overview of the research landscape, it is recommended that the authors include a table summarizing the work of other researchers in this area. This table should compare and contrast the key findings, methods, and materials used in similar studies, allowing readers to easily visualize the contributions and advancements in the field.
Response 14. A summary of the results for CMC-based SAP production is presented in Table 3.
Point 15. Begin your conclusion section with a summary of your work, emphasizing the main findings and the gap your work fills in the literature. Conclude your work in this section as well. Current version only summarizes your work without concluding
Response 15. We have modified the conclusion according to your suggestion.
Thanks for your comments. You help us become better!
Reviewer 3 Report
Comments and Suggestions for Authors
Lavlinskaya et al. reported the paper entitled Enhancement of Water Uptake in Composite Superabsorbents Based on Carboxymethyl Cellulose through Porogen Incorporation and Lyophilization. The paper lacks of novelty. The data is general. There is not much data to meet the journal quality. The paper needs considerable revision. Therefore, I suggest rejecting and resubmitting.
1. The authors are requested to add a detailed paragraph about SAP with more representative examples.
2. The authors have requested the advantages and disadvantages of SAP.
3. The authors should discuss different types of cross-linkers and the comparison between the cross-linkers.
4. The figure quality could be better. The authors are requested to present the data in a representable way. Please only mark the important wavenumbers in the FTIR spectrum.
5. There needs to be more than SEM images. The authors are requested to add AFM and TEM data to support the hypothesis.
6. Please put a schematic.
7. The BET experiment needs to be included. Please provide the data.
8. DSC and Compression tests are missing.
9. Water absorption kinetics and re-swelling capacity experiments must still be included.
Author Response
We would like to thank the Reviewer for careful reading of our manuscript. In the view of the constructive criticism by the Reviewer, we have revised the manuscript considerably. All our corrections are highlighted in yellow in the text.
Lavlinskaya et al. reported the paper entitled Enhancement of Water Uptake in Composite Superabsorbents Based on Carboxymethyl Cellulose through Porogen Incorporation and Lyophilization. The paper lacks of novelty. The data is general. There is not much data to meet the journal quality. The paper needs considerable revision. Therefore, I suggest rejecting and resubmitting.
Point 1. The authors are requested to add a detailed paragraph about SAP with more representative examples.
Response1. We have made modifications to the introduction based on your suggestions.
Point 2. The authors have requested the advantages and disadvantages of SAP.
Response 2. We have revised the introduction based on your suggestions.
Point 3. The authors should discuss different types of cross-linkers and the comparison between the cross-linkers.
Response 3. We have added a paragraph in the introduction discussing the influence of cross-linkers on water-absorbing capacity. We hope you find it suitable.
Point 4. The figure quality could be better. The authors are requested to present the data in a representable way. Please only mark the important wavenumbers in the FTIR spectrum.
Response 4. All figures in better quality have been uploaded with the manuscript. Fig. 1 (FTIR spectrum) has been modified according to your comment.
Point 5. There needs to be more than SEM images. The authors are requested to add AFM and TEM data to support the hypothesis.
Response 5. Thank you for your valuable comment. Indeed, the availability of various methods allows us to highlight the depth of the study and comprehensively support the hypothesis being tested. Unfortunately, not all desired methods are always accessible to the researcher or suitable for the object being studied. The AFM method available to us can only scan an area of up to 10x10 μm, which is significantly smaller than the pore size obtained in this work. As a result, the findings from this technique are not as significant or relevant for this study. Additionally, using TEM to obtain informative images is challenging, as it is difficult to select particles of a size that both contain a porous structure and are accessible to the electron beam. Therefore, we do not recommend this method for studying macroporous objects.
Point 6. Please put a schematic.
Response 6. The synthesis flowchart has been added as Scheme 1.
Point 7. The BET experiment needs to be included. Please provide the data.
Response 7. Thank you for your insightful comment. Indeed, when studying a porous object, it is valuable to examine the size and distribution of pores using the BET method. However, most devices for this experiment are designed to work accurately with pores smaller than 500 nm. In our study, the pore diameter exceeds several microns, making the application of this method challenging. We attempted to conduct this analysis, but the results were not reliable, so they are not presented in this manuscript. Instead, we have included porosity data obtained through ethanol displacement.
Point 8. DSC and Compression tests are missing.
Response 8. DSC data have been added as Figure S1. Unfortunately, we do not have the equipment available to perform compression tests. This manuscript is a part of continuous research, so we will try to perform compression test in our future studies.
Point 9. Water absorption kinetics and re-swelling capacity experiments must still be included.
Response 9. Water absorption kinetics are presented in Fig. 5a, while re-swelling data are shown in Fig. 6.
Thanks for your comments. You help us become better!
Round 2
Reviewer 2 Report
Comments and Suggestions for Authors
Authors have significantly improves the quaility of the manuscript by incorporating the suggested changes as well as provided comprehensive reply to the comments. This manuscript may be accepted in the current state.
Reviewer 3 Report
Comments and Suggestions for Authors
The authors have tried to justify the answers I raised. The authors have included many portions as requested. The manuscript can be accepted in its current form.